# Single-molecule imaging reveals the concerted release of myosin from regulated thin filaments

**Quentin M Smith[1†], Alessio V Inchingolo[1†], Madalina-Daniela Mihailescu[2], Hongsheng Dai[2], Neil M Kad[1]\***

[1]School of Biosciences, University of Kent, Canterbury, United Kingdom; [2]Department of Mathematical Sciences, University of Essex, Colchester, United Kingdom

**Abstract** Regulated thin filaments (RTFs) tightly control striated muscle contraction through calcium binding to troponin, which enables tropomyosin to expose myosin-binding sites on actin. Myosin binding holds tropomyosin in an open position, exposing more myosin-binding sites on actin, leading to cooperative activation. At lower calcium levels, troponin and tropomyosin turn off the thin filament; however, this is antagonised by the high local concentration of myosin, questioning how the thin filament relaxes. To provide molecular details of deactivation, we used single-molecule imaging of green fluorescent protein (GFP)-tagged myosin-S1 (S1-GFP) to follow the activation of RTF tightropes. In sub-maximal activation conditions, RTFs are not fully active, enabling direct observation of deactivation in real time. We observed that myosin binding occurs in a stochastic step-wise fashion; however, an unexpectedly large probability of multiple contemporaneous detachments is observed. This suggests that deactivation of the thin filament is a coordinated active process.

**\*For correspondence:**
n.kad@kent.ac.uk

[†]These authors contributed equally to this work

**Competing interest:** The authors declare that no competing interests exist.

## Introduction

Striated muscle contraction is mediated by the interaction of myosin II with actin and is governed by the calcium concentration within the myocyte. In the sarcomere, myosin II is assembled into thick filaments, which cyclically bind to the actin-containing thin filaments in the presence of adenosine triphosphate (ATP) to generate force. However, access to actin is modulated by the control proteins troponin (Tn) and tropomyosin (Tm). Tn is a complex of three proteins that interact with both actin and the 40 -nm-long filamentous protein Tm. Calcium binding triggers a conformational change in Tn that permits Tm to slide across actin (*Poole et al., 2006*). This movement exposes sites on actin for myosin to bind, which in turn leads to the further exposure of myosin-binding sites on actin (*Gordon et al., 2000*). Together, these interactions result in cooperative activation of the thin filament and the generation of force. A further level of force control is provided by the thick filament, which responds to environmental cues to alter the amount of available myosin (*Linari et al., 2015*; *Reconditi et al., 2017*; *Piazzesi et al., 2007*). How muscle deactivates is as important as its activation, and defects in this process have been implicated in cardiac disease (*Tardiff, 2005*).

Activation of the thin filament is hypothesised to be modulated by the accessibility of myosin-binding sites on actin within the context of a three-state mechanism (*McKillop and Geeves, 1993*). In the first of these states (blocked), tropomyosin sterically impedes myosin from binding to actin. Upon calcium binding to troponin, tropomyosin shifts across actin, partially uncovering myosin-binding sites, allowing a weak interaction between myosin and actin (closed state). Finally, in the open state, myosin binds strongly to actin and can generate force. Due to the stiffness of Tm, its movement to the open state imposed by myosin head binding also uncovers further myosin-binding sites on actin, leading to cooperative activation (*Walcott and Kad, 2015*). Since actin is a long filamentous molecule, there

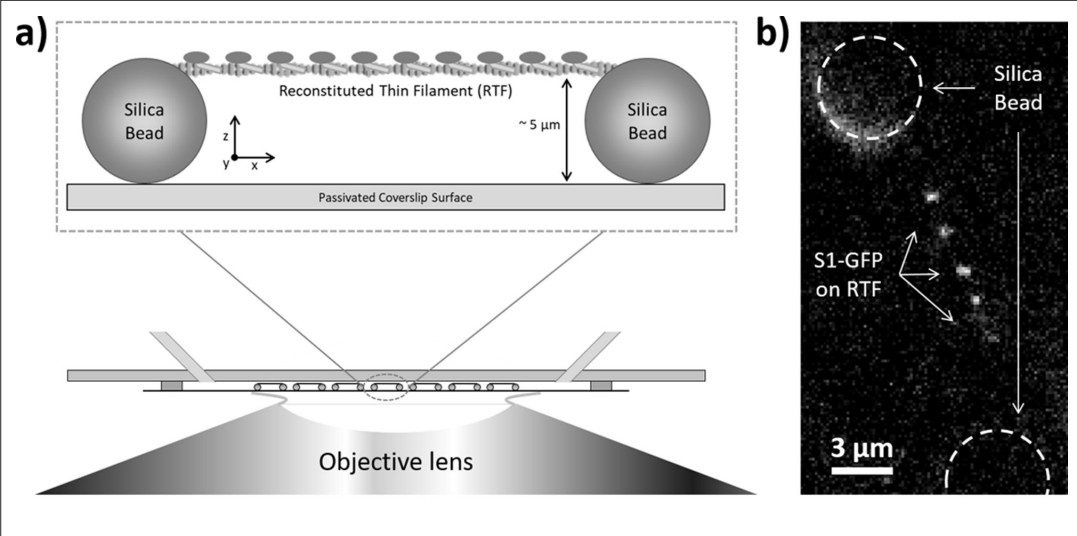

**Figure 1.** Imaging individual GFP-tagged myosins interacting with regulated thin filaments. (**a**) Regulated thin filaments (RTFs) are suspended between surface-immobilised beads (top) using a microfluidic chamber (bottom). The silica beads are functionalised with poly-L-lysine that adheres them to the passivated (methoxypolyethylene glycol (mPEG)-coated) coverslip surface and to the RTFs. Illumination is achieved at an oblique angle to reduce background, and fluorescence detection occurs through the objective lens. (**b**) Top-down view of GFP-tagged myosin-S1 (S1-GFP) molecules bound to an RTF suspended between silica beads (dashed circles). Each pixel is 126.4 × 126.4 nm.

is no clear structural definition of the cooperative unit size. Instead, this size is defined functionally in terms of the amount of actin exposed for myosin to bind (*Maytum et al., 1999*; *Desai et al., 2015*; *Geeves and Lehrer, 1994*). These and other studies (*Heeley et al., 2002*; *Heeley et al., 2006*) suggest that myosin binding to a thin filament is capable of holding Tm open for the association of up to 7–14 further myosins; indeed, imaging studies have revealed the potential for much longer active stretches (*Vibert et al., 1997*; *Marston, 2003*). In reverse, this process is less clear. How does Tm return to the blocked state in the presence of myosins at a very high local concentration (>0.15 mM; *Ferenczi et al., 1984*) as the calcium concentration drops?

Using a single-molecule approach to measure the binding and release of fluorescent myosin molecules to thin filaments in sub-maximal calcium conditions, we are able to shed light on the process of deactivation. In this assay (*Desai et al., 2015*), single thin filaments are suspended between surface-immobilised beads to enable access of myosin to actin unimpeded by a surface. With sufficient myosin, at the mid-point of calcium-induced activation (pCa$_{50}$), a metastable activation process is observed. Myosin binds to regions of actin for long enough to permit additional myosins to bind in close proximity, without favouring complete propagation of activation as seen at saturating calcium (*Desai et al., 2015*). Here, we temporally resolve the interactions of fluorescently tagged myosin S1 with the thin filament. Changes in fluorescence intensity within active regions correspond to the release or binding of myosins. By following these events, we can statistically reconstruct the fate of active regions. We found that, as expected, myosin binds to thin filaments stochastically and forms clusters. However, a much-larger-than-expected probability for the simultaneous release of all myosins within an active cluster was observed in the data. In the metastable conditions studied, this was the most probable outcome for an active region. This highly elevated collapse probability suggests a concerted mechanism of deactivation (relaxation) and explains the ability of muscle to relax in conditions that would be expected to still permit myosin binding.

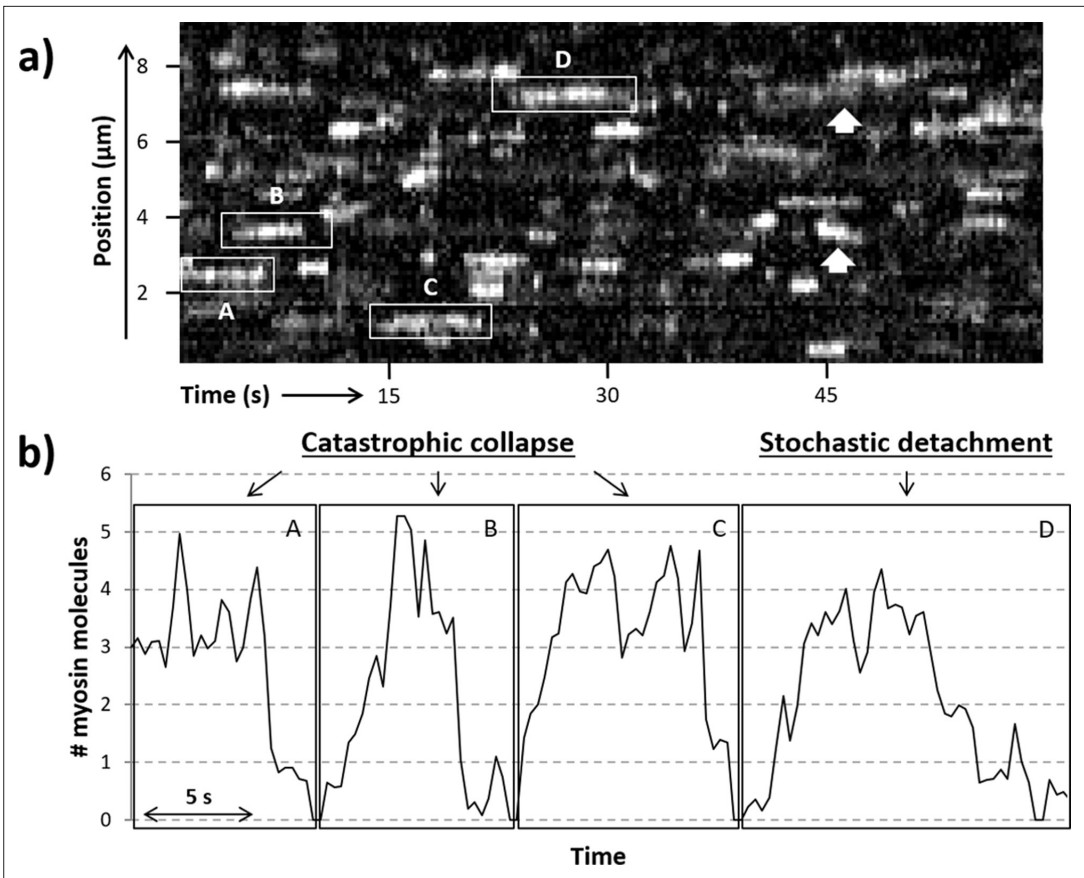

**Figure 2.** Active regions on an RTF tightrope. (**a**) A representative kymograph of a regulated thin filament (RTF) in sub-maximal activating conditions, with some of the locally active regions (highlighted in boxes) labelled A to D. (**b**) The number of GFP-tagged myosin-S1 (S1-GFP) molecules bound in the highlighted active regions is shown. Collapse of the active region is seen in particular for areas A, B, and C, whereas region D shows both stepwise attachments and detachments. White arrows highlight example regions where the active cluster is seen to move along the thin filament. Data were obtained for 5 nM S1-GFP at pCa 6 and 0.1 μM adenosine triphosphate (ATP).

## Results

### Generation and qualitative assessment of metastable acto-myosin interactions

Single-molecule imaging of green fluorescent protein (GFP)-tagged myosin-S1 (S1-GFP) loading and release from regulated thin filaments (RTFs) offers a direct spatial and temporal view of how activation occurs. In our previous work (*Desai et al., 2015*), we developed a new approach to follow this process on RTF tightropes. These structures consist of individual RTFs suspended between beads adhered to a microscope coverslip surface (*Figure 1a*). The RTFs are constructed using microfluidics, which are also used to add the assay components. In the presence of S1-GFP, we are able to detect binding to the RTF using fluorescence detection of the GFP; only those molecules binding to the tightrope are stationary long enough to register as a signal (*Figure 1b*).

Previously, S1-GFP was seen to bind in clusters; the occupancy of these active regions was dependent on the concentration of myosin, ATP, and calcium in the steady state. Only thin filaments, not bare actin, showed this clustering behaviour, indicating this was a process regulated by the accessory proteins, troponin and tropomyosin. Here, using the same experimental approach, we measured the dynamics of S1-GFP binding to and releasing from active regions. To increase the occurrence of formation and loss of active regions on the RTF, we tuned the [ATP] to prolong attachment while lowering the calcium concentration to near the $pCa_{50}$ (mid-point of activation; *Desai et al., 2015*). This sub-maximal activation of the RTF prevents it from being fully turned on, leading to the formation of metastable active regions (*Figure 2a*). Examination of these metastable active regions indicates that

clusters of S1-GFP are formed that can move and spread laterally along the thin filament (*Figure 2a*, arrows; see also *Desai et al., 2015*) with no directional bias, as expected from a stochastic process. This indicates that the binding/release of S1-GFP occurs stochastically with equal probability at both ends of the cluster (*Desai et al., 2015*). Since the intensity within the active region is directly related to the number of myosins present, we can time-resolve the loading and release of myosin. In active regions, we observed that binding occurs predominantly stepwise (*Figure 2b*), whereas detachment occurs both stepwise or through contemporaneous detachment of multiple myosin molecules. This qualitative assessment of the data prompted a more robust examination to determine the process of detachment.

## Quantifying the probability of transitioning from one state to another

To understand the probability of S1-GFP leaving from or associating with an active region, we calculated the flux for each active region size and then aggregated the data to construct transition tables. The position and intensity of every active region were monitored using an automated approach enabling unbiased detection of the number of myosins in every active region. This approach was corroborated using an alternative statistical analysis (see 'RJMCMC' section of 'Materials and methods').

Movies of S1-GFP binding to and releasing from thin filament tightropes were converted into kymographs along the contour of actin using ImageJ (*Figure 3a and b*). The dosage of laser illumination, which is a product of intensity and duration, did not change interaction lifetimes of individual myosins, indicating that photobleaching was not affecting this study in these conditions (*Desai et al., 2015*). Any systematic asymmetries in background noise were removed from the kymographs using ImageJ rolling-ball background subtraction (radius = 50 pixels). Extraction of fluorescence intensities from the kymographs was performed using an automated MATLAB routine that fits each time-slice (corresponding to a frame of the movie) to an autonomously determined number of Gaussian distributions

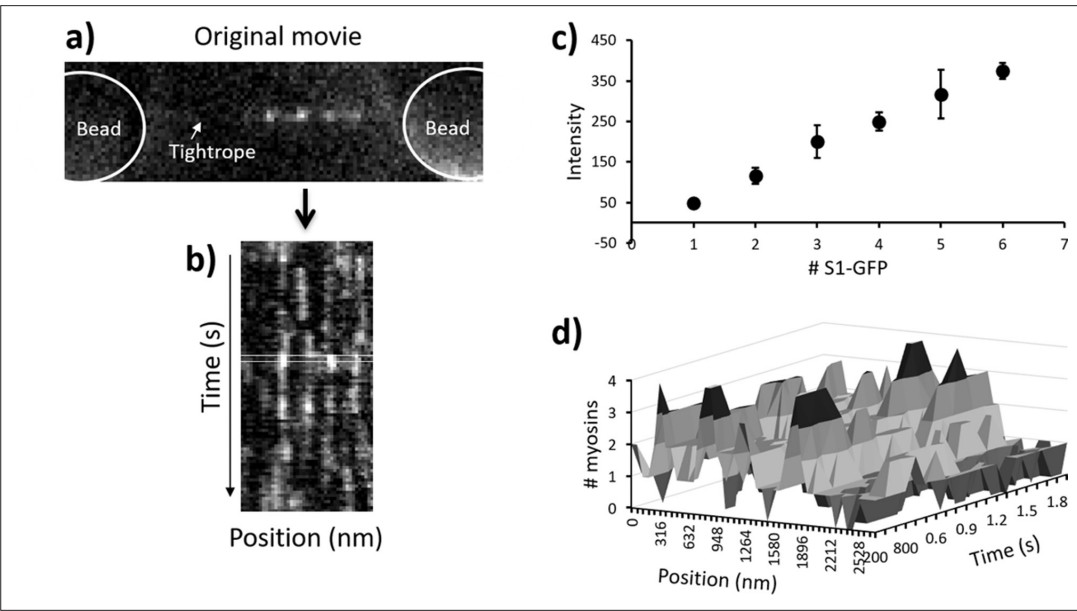

**Figure 3.** Pipeline for semi-automated data analysis. (**a**) A representative snapshot from a metastable regulated thin filament (RTF), the beads being greyed out to highlight the active regions. (**b**) This region is transformed into a kymograph where vertical streaks indicate active regions. During analysis, a running window of one frame (rectangular box) is moved across the data through time, and each line profile of that box is fitted to a sum of multiple Gaussians using MATLAB. All fitted intensity values are plotted as a histogram and fitted again to a sum of Gaussians (see *Desai et al., 2015*). (**c**) The resulting mean intensity values are plotted to reveal a straight line ($R^2$ = 0.999); the error bars represent 2 x SD for the Gaussian fit. The mean ± 2 x SD represents the range of intensity values used to define the presence of that specific number of myosins. These are used to rescale the kymograph into the number of myosin molecules present. (**d**) A section of the kymograph in (**b**) rescaled to the number of myosins. Subsequent analysis is used to define the myosins that belong to a specific active region (see main text). The data shown here were obtained with 10 nM GFP-tagged myosin-S1 (S1-GFP) at pCa 6 and 0.1 µM adenosine triphosphate (ATP).

(*Figure 3b*, and the code is available at https://github.com/Kad-Lab/catastrophic_collapse; *Smith, 2021* copy archived at swh:1:rev:5137e03d2062938c8d1ff15b7de5a97f5740b05f). Signals were extracted following noise removal by thresholding; therefore, it is possible that some single bound myosins may have been lost. However, *Figure 3c* shows a close extrapolation to zero, indicating this effect is minor. The binding and release of myosin during periods of metastable activity leads to a shift in the mean position of the active region over time. Therefore, it was necessary to assign fluorescence peaks to active regions. This was achieved by firstly processing the results from the MATLAB routine to extract two features: the fluorescence intensity and position. Using Microsoft Excel's Solver Add-in, the intensities of all myosin peaks were fitted to multiple Gaussians (*Desai et al., 2015*), and then the peak intensities were plotted against the number of binders for each kymograph (*Figure 3c*). This was used to convert the fluorescence intensity into numbers of myosins; the error bars in *Figure 3c* represent 2 x SD of the Gaussian fits, which is the range used to assign the intensity to that particular number of bound myosins. This method exhibits minimal overlap of fluorescence integrals and therefore leads to the accurate assignment of intensity to the number of myosins. Using only the position of the fluorescent spot and the corresponding intensity value, we recreated an idealised movie for tracking analysis to enable automated assignment (threading) of active region positions. Each frame in this idealised movie corresponds to a time-slice in the kymograph. We repurposed the automated FIJI plugin, Trackmate, to track the position of spots (*Tinevez et al., 2017*) and link together fluorescent signals corresponding to a single active region.

Knowing the number of myosins in each active region at every point in time enabled us to calculate transition matrices, which are tables that indicate what happens when the occupancy of an active region changes. To generate a transition matrix, we examined the fate of each active region (also termed cluster) by measuring the frequency of transitions from one cluster size to another from one original video frame to the next. From these measurements, we generated a matrix of probabilities (*Figure 4a*). The columns represent the final cluster size and the rows, the starting size. The central diagonal is zero, since we were measuring movement away from the current cluster size. Analysis was limited to cluster sizes of 6 due to limitations in the number of measured large clusters; therefore, the very few larger clusters were discarded from the analysis. All rows sum to 1, and there are no vertical transitions. For example, starting with a cluster size of four molecules in *Figure 4a*, the probability of releasing an S1-GFP is 20 %, whereas the probability of a single S1-GFP binding is 5 %. It can be seen that near the central diagonal, there is an increased probability, corresponding to the release/binding of single S1-GFPs. However, larger transition probabilities are seen in the first column, indicating a high propensity for complete detachment of all molecules in a single frame.

Activation occurs through the initial association of myosin to open the thin filament for subsequent myosins to bind (*Desai et al., 2015*). These bind in a diffusion-limited process and, although stochastic, the process of activation is predictable. Likewise, the release of myosin from the thin filament is predicted from the adenosine diphosphate (ADP)-release rate constant and the second-order ATP-binding rate constant. To determine the association probability empirically, we used the information embedded in the transition matrix shown in *Figure 4a*. Because of the stochastic nature of myosin binding, we used the mean probability for the first binder (the first value to the right of the zero diagonal in *Figure 4a*) as an estimate of the association probability. The probability of association will scale according to the power of the number of attached molecules. Therefore, if two myosins bind, the expected probability for this event is square that of a single binder, and for three myosins, it is the cubed probability and so on. Using this analysis, it is possible to calculate the expected probability for any number of myosins binding in one movie frame to an existing active region, shown in *Figure 4b*, as the values to the right of the diagonal. The probability of detachment is calculated from the parameters that govern attachment time, ADP release, and ATP binding. ADP release for skeletal muscle myosin is extremely fast in comparison with ATP binding at 0.1 µM ATP; therefore, we only calculated the probability of detachment from the latter. Using a second-order ATP-binding rate constant of 1.9 µM s$^{-1}$ (*Desai et al., 2015*; *Geeves and Lehrer, 1994*), the detachment rate constant is calculated as 0.19 s$^{-1}$. The duration of each frame is 300 ms; therefore, the expected probability of detachment during a single frame is $(1-e^{(-0.19*0.30)}) = 5.6$ %. As with attachment, this value will scale according to the number of myosins in a cluster; however, any molecule can leave the cluster and, therefore, the probability of detachment becomes the product of a single detachment and the active region size, scaled by a term that takes into account the possible combinations of myosins detaching from a cluster (see

**a) Final Cluster Size - Regulated Thin Filament**

| Start Cluster Size | 0 | 1 | 2 | 3 | 4 | 5 | 6 | Legend |
|---|---|---|---|---|---|---|---|---|
| 1 | 0.30 | 0.00 | 0.40 | 0.19 | 0.06 | 0.02 | 0.02 | 0.00 |
| 2 | 0.37 | 0.33 | 0.00 | 0.18 | 0.09 | 0.02 | 0.01 | 0.20 |
| 3 | 0.30 | 0.25 | 0.31 | 0.00 | 0.07 | 0.04 | 0.04 | 0.40 |
| 4 | 0.20 | 0.18 | 0.34 | 0.20 | 0.00 | 0.05 | 0.02 | 0.60 |
| 5 | 0.25 | 0.20 | 0.22 | 0.20 | 0.10 | 0.00 | 0.03 | 0.80 |
| 6 | 0.27 | 0.23 | 0.11 | 0.29 | 0.04 | 0.05 | 0.00 | 1.00 |

**b) Final Cluster Size - Predicted**

| Start Cluster Size | 0 | 1 | 2 | 3 | 4 | 5 | 6 | Legend |
|---|---|---|---|---|---|---|---|---|
| 1 | 0.24 | 0.00 | 0.65 | 0.10 | 0.01 | 0.00 | 0.00 | 0.00 |
| 2 | 0.01 | 0.39 | 0.00 | 0.51 | 0.08 | 0.01 | 0.00 | 0.20 |
| 3 | 0.00 | 0.03 | 0.48 | 0.00 | 0.42 | 0.06 | 0.01 | 0.40 |
| 4 | 0.00 | 0.00 | 0.04 | 0.54 | 0.00 | 0.36 | 0.05 | 0.60 |
| 5 | 0.00 | 0.00 | 0.00 | 0.07 | 0.61 | 0.00 | 0.32 | 0.80 |
| 6 | 0.00 | 0.00 | 0.00 | 0.01 | 0.12 | 0.87 | 0.00 | 1.00 |

**c) Final Cluster Size - Tropomyosin (only) Thin Filament**

| Start Cluster Size | 0 | 1 | 2 | 3 | 4 | 5 | 6 | Legend |
|---|---|---|---|---|---|---|---|---|
| 1 | 0.39 | 0.00 | 0.31 | 0.13 | 0.10 | 0.03 | 0.03 | 0.00 |
| 2 | 0.34 | 0.32 | 0.00 | 0.15 | 0.11 | 0.04 | 0.04 | 0.20 |
| 3 | 0.30 | 0.25 | 0.28 | 0.00 | 0.12 | 0.02 | 0.04 | 0.40 |
| 4 | 0.30 | 0.25 | 0.21 | 0.15 | 0.00 | 0.03 | 0.05 | 0.60 |
| 5 | 0.29 | 0.27 | 0.25 | 0.12 | 0.04 | 0.01 | 0.02 | 0.80 |
| 6 | 0.30 | 0.23 | 0.22 | 0.14 | 0.07 | 0.04 | 0.00 | 1.00 |

**Figure 4.** Predicted and measured transitions between cluster sizes. By measuring the number of myosins in each cluster, the transition rates between them can be determined. (**a**) Using the transition rates measured for regulated thin filaments (RTFs), a transition probability table was constructed, where the rows are the starting cluster size and the columns, the final. The central diagonal is zero because this table measures the probability of leaving that cluster; all numbers to the right of the diagonal are binding events, and to the left are detachments. Data were obtained at 5 and 10 nM GFP-tagged myosin-S1 (S1-GFP), pCa 6, and 0.1 µM adenosine triphosphate (ATP) and pooled for analysis. A total of 23 kymographs were used to yield 8140 transitions for this analysis. (**b**) Using the probability of association empirically determined from (**a**) and known kinetic parameters that govern myosin release, it is possible to construct an expected table of transition probabilities (see 'Discussion' for more details). The overall pattern of behaviour between measured (**a**) and predicted (**b**) transition probabilities is similar to the right of the central diagonal; however, to the left, there is increased probability of myosin release. (**c**) The transition probabilities for tropomyosin alone follow a similar pattern to that of the regulated thin filament, indicating that the accelerated release of myosin is mediated mostly by tropomyosin. Data were obtained at 5, 15, and 20 nM S1-GFP in 0.1 µM ATP and pooled for analysis. A total of 19 kymographs were used to yield 2332 transitions for this analysis. Data are available in *Figure 4—source data 1*.

The online version of this article includes the following source data and figure supplement(s) for figure 4:

**Source data 1.** Excel spreadsheet of compiled data used for *Figure 4a–c*.

**Figure supplement 1.** RTF transition table measured using RJMCMC.

**Figure supplement 1—source data 1.** Excel spreadsheet containing source data for *Figure 4—figure supplement 1*.

'Data analysis' section of 'Materials and methods'). Using these conditions, we created the values to the left of the central diagonal to complete the predicted table of transition probabilities (*Figure 4b*). Neither the top right nor the bottom left of the predicted value table is populated. This strongly contrasts with the transition table for measured RTFs, which shows a substantial population in the bottom left, indicating that myosin detachment is coordinated.

Towards understanding the mechanism of concerted detachment of myosins, we studied the attachment/detachment probabilities of S1-GFP for actin-decorated tropomyosin alone. Tropomyosin is known to impede myosin attachment as evidenced by inhibition of in vitro motility (*VanBuren et al., 1999*) and the S1 ATPase (*Lehrer and Geeves, 1998*). However, as with motility, it is possible to overcome this inhibition through strong binding activation (*Lehrer and Geeves, 1998*), alone tropomyosin is thought to occupy the closed state (*Poole et al., 2006*; *Geeves, 2012*), which can move to the open (M) state (*Orzechowski et al., 2014*). Therefore, as expected, in this assay, we noted a requirement for at least 5 nM S1-GFP before metastability is established (*Figure 4c*). This contrasts with actin alone, which shows substantial and randomly located decoration with myosin at only 1 nM S1-GFP (*Desai et al., 2015*). By performing the same analysis as above but with tropomyosin alone, we generated a similar transition matrix to that of fully reconstituted thin filaments in metastable conditions (*Figure 4c*). Myosin binding once again appeared stochastic and detachment concerted with Tm alone. We were unable to perform a similar analysis on undecorated actin since active clusters were not formed; instead, decoration increased across the filaments until the actin appeared fully decorated (*Desai et al., 2015*).

## Discussion

Classical cooperative systems use the energy of substrate binding to activate association of further substrates, all within the confines of the structural unit. These confines determine the number of substrate molecules that can bind (*Monod et al., 1965*; *Koshland et al., 1966*; *Cliff et al., 1999*). The sarcomeric thin filament presents a uniquely cooperative environment, where the number of substrate molecules, in this case myosin, that can associate is not structurally defined. Furthermore, this system is allosterically controlled through calcium binding to troponin which, together with tropomyosin, controls myosin's access to binding sites on actin (*Gordon et al., 2000*). Myosin binding results in a more classical cooperative activation as it holds Tm away from adjacent sites, thus propagating further myosin association (*Desai et al., 2015*; *Craig and Lehman, 2001*; *Trybus and Taylor, 1980*; *Swartz and Moss, 1992*; *Kad et al., 2005*). Despite this understanding, many crucial details remain to be understood, including how the thin filament relaxes. This process occurs as the calcium concentration drops in the myocyte, leading to actin re-binding of troponin I and repression of myosin binding (*Galinska-Rakoczy et al., 2008*). Given the very high local concentrations of myosin in the sarcomere (upwards of 0.15 mM; *Ferenczi et al., 1984*), troponin I binding and Tm movement will compete with myosin re-binding, making relaxation difficult. To reveal this mechanism, we have studied the dynamic process of metastable myosin binding that occurs near the mid-point of the calcium activation curve. In these conditions, we directly observed single molecules of fluorescently tagged myosin S1 interacting with the thin filament, forming clustered regions of activation. Detailed examination of the binding and release of myosins revealed an unexpectedly high probability of concerted myosin release, suggesting an active process of myosin detachment that we term catastrophic collapse.

### Myosin detachment from active regions is cooperative

How muscle relaxes has been studied for over a century (*Hartree and Hill, 1921*) using a variety of increasingly sophisticated approaches (for an excellent overview, see *Poggesi et al., 2005*). The precise molecular basis of relaxation, where the time-course is comparable to that of contraction (*Jewell and Wilkie, 1960*), is still uncertain. However, a series of in vitro studies have indicated that a time lag exists between calcium drop and myosin detachment (*Geeves and Lehrer, 1994*), possibly explaining the shoulder of relaxation (*Brunello et al., 2009*; *Huxley and Simmons, 1970*). By working in a metastable condition that exists near the $pCa_{50}$, we were able to visualize the release of myosins directly, with the limitation that a direct correlation with local calcium release is not possible. Our temporal resolution during imaging (3.3 fps) was sufficient to detect stepwise detachment based on myosin's expected attached lifetime (see above), enabling the construction of a predicted transition matrix (*Figure 4b*). This revealed an increasingly lower probability of dis/association further away from the central diagonal, consistent with a stochastic model where active regions increase or decrease in size primarily in single steps. However, the observed transition rates from our measured data reveal a striking difference from the predicted behaviour. A comparison of the predicted versus the measured probabilities for the first column of the transition matrices provides a stark contrast (*Figure 5*). The

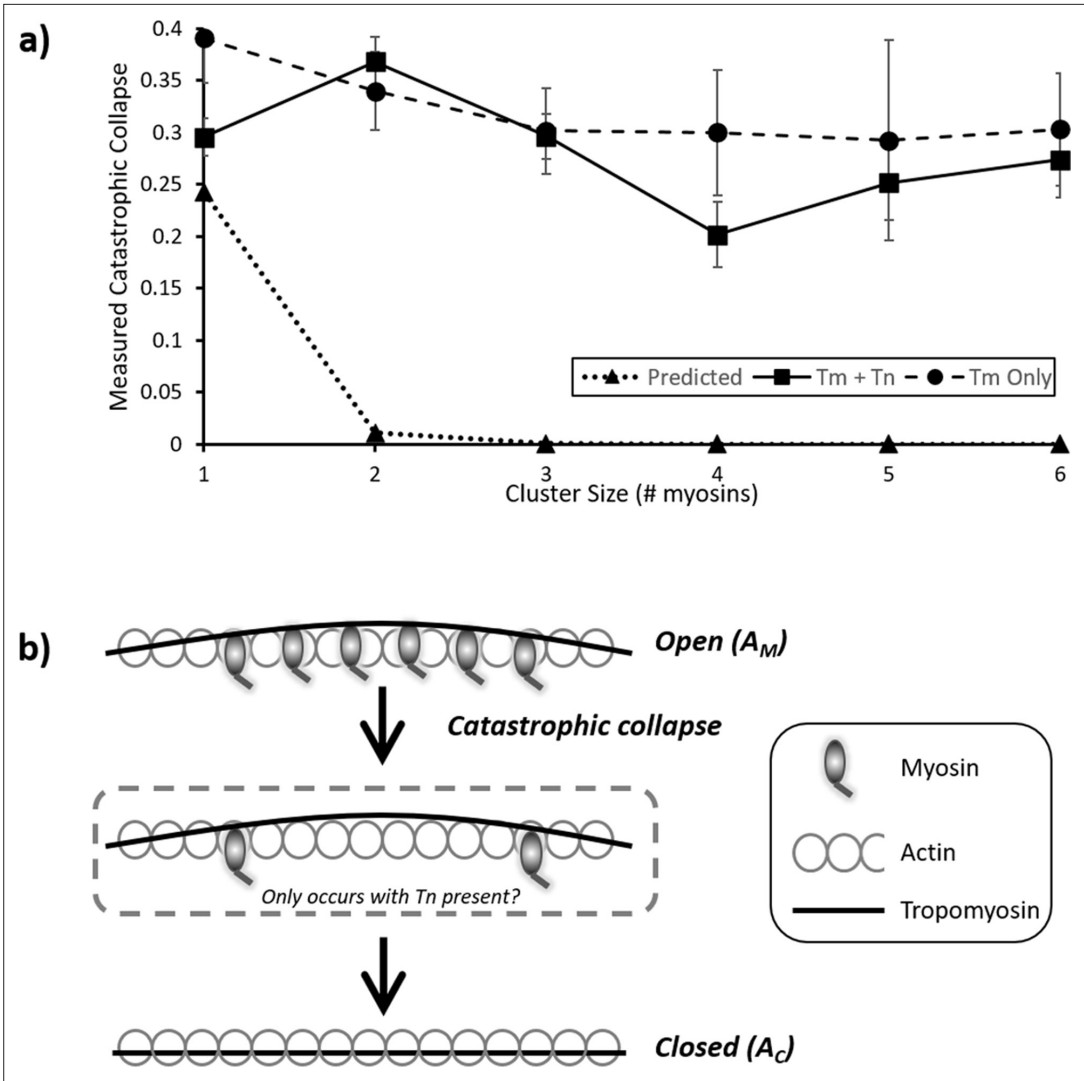

**Figure 5.** Probability and mechanism of catastrophic collapse. (**a**) Plots of the values from the first columns of each table in *Figure 4*; this provides the probability of complete collapse. For the predicted data (triangles), an expected power law dependence correlates with cluster size is seen. However, both for the measured RTF data (squares) and tropomyosin only (circles) a much higher probability of collapse is observed. For RTFs this peaks at two and decreases marginally with larger clusters whereas Tm only shows a steady decline. Error bars represent SEM and were calculated from the values in the underlying kymographs, the number of observations is the same as for *Figure 4*. Data is available in *Figure 5—source data 1*. (**b**) Cartoon representation of the process of catastrophic collapse. Several myosins bind to actin holding tropomyosin in an open ($A_M$) configuration (top). However, all myosins release in one step allowing the tropomyosin to relax into the closed ($A_C$) state (bottom). In the presence of troponin there may be a transitory state with two myosins bound (middle).

The online version of this article includes the following figure supplement(s) for figure 5:

**Source data 1.** Excel spreadsheet of compiled data used for *Figure 5a*.

expected detachment of all myosins in a single step decreases logarithmically with increasing active region size, but the measured probability appears to only marginally decline. This corroborates the observed large decreases in the raw data intensity of active regions (*Figures 2 and 3*), indicating several myosins leaving contemporaneously. The stochastic nature of photobleaching cannot explain this synchronous reduction in fluorescence. Therefore, these data indicate that myosin release from thin filaments is concerted. As a consequence, the detachment of myosin must be accelerated by tropomyosin.

Interestingly, the data in *Figure 4a* potentially also show a propensity for a favoured intermediate on the way to detachment of two bound myosins. Regardless of the initial cluster size, there is an enhanced probability that collapse will result in two bound myosins. This emerges from the independent data analysis, and the higher value for two myosins detaching to zero suggests that the detachment to two myosins could be intermediary to complete collapse. Furthermore, we were able to convert the probabilities of complete detachment shown in *Figure 5a* back into detachment rate constants by mathematically reversing the process we used to construct the expected probability matrix (*Figure 4b* and 'Materials and methods'). From this, we could predict an increase in detachment rate constant from ~3 s$^{-1}$ to ~6 s$^{-1}$ as the cluster size increased from 2 to 6 in the presence of troponin and tropomyosin. Further investigations using super-resolution techniques would shed light on both of these intriguing aspects revealed by the experiments performed here.

The precise mechanism of how catastrophic collapse occurs on the thin filaments is not completely revealed in these experiments although our data suggest troponin is not required. This is surprising, given the large structural footprint of troponin (*Yamada et al., 2020*; *Tobacman, 2021*); however, the observation that tropomyosin alone is capable of modulating catastrophic collapse and relaxation is supported by previous studies of phosphorylated Tm (*Rao et al., 2011*; *Nixon et al., 2013*). There were differences in the collapse behaviour in the absence of troponin; firstly, the probability of collapse was greater across all cluster sizes except for cluster size 2. This suggests that Tm ejects myosin from the actin filament more easily than in the presence of Tn. The mechanism by which troponin enhances activation of the thin filament was revealed by experiments previously performed using an actin mutation that enhances the blocking capability of Tm alone (D292V; *Rynkiewicz et al., 2017*). Only in the presence of troponin and calcium is the blockade released, suggesting that Tn pulls Tm away from myosin-binding sites, changing the energy landscape of the actin filament (*Orzechowski et al., 2014*; *Ali et al., 2010*; *Lehman et al., 2009*; *Perz-Edwards et al., 2011*). This view of activation is relevant to the observation here that Tm is more capable of releasing myosins, because troponin (and calcium) is not present to reduce the energetic propensity of Tm to enter the blocked state. This is also corroborated by the measured reduction in cooperative unit size for Tm, compared with Tm- and Tn-decorated actin (*Geeves and Lehrer, 1994*). Finally, it is relevant to speculate how myosin is released; is it pushed off from actin or instead released by conformational capture? The process of detachment in a metastably active thin filament would revolve around the local presence of calcium and how its loss mediates detachment. It seems unlikely that myosin would be physically pushed off actin given the energy required; however, since Tm is a very long molecule, any change in affinity for actin would be magnified by the power of the number of actin monomers involved (*Ali et al., 2010*). An alternative mechanism for release would involve micro-dissociation events that permit tropomyosin to 'invade' myosin's binding site, facilitating release. Such a process has been shown for DNA strand displacement (*Simmel et al., 2019*) and also for proteins that exchange on DNA (*Graham et al., 2011*). These observations from orthogonal fields may inform, at the sub-molecular scale, how myosin can be displaced from actin by tropomyosin. Indeed, recent structural studies suggest that loop 4 of myosin is the crux of the revolving interactions between myosin and tropomyosin (*Doran et al., 2020*), suggesting an excellent target for future investigations.

## Consequences of accelerated detachment

Tropomyosin movement on the surface of actin is either rapid, on the order of, or slightly slower than the rate of myosin head release (*Geeves and Lehrer, 1994*; *Trybus and Taylor, 1980*; *Tesi et al., 2002*). When under load, the release of myosin is slowed and, therefore, tropomyosin may be capable of assisting the release of myosins, offering a possible molecular explanation for chaotic relaxation of isometric myofibrils (*Brunello et al., 2009*). Such a role would not be as significant during rapid shortening when both the release rate of heads is very high and the number of attached heads is estimated to be low (<10%; *Piazzesi et al., 2007*).

We have detected active regions of up to nine bound myosins (due to the smaller sample sizes for these larger regions, only those up to six bound myosins were analysed in this study), consistent with previous studies (*Desai et al., 2015*; *Geeves and Lehrer, 1994*). The spatial range of this activation is not clear from the studies here because the fluorescence overlap prevents clear super-resolution of the binding locations. Structural and functional studies have suggested that the active region can be very long (*Desai et al., 2015*; *Vibert et al., 1997*; *Marston, 2003*); so in the sarcomere with very

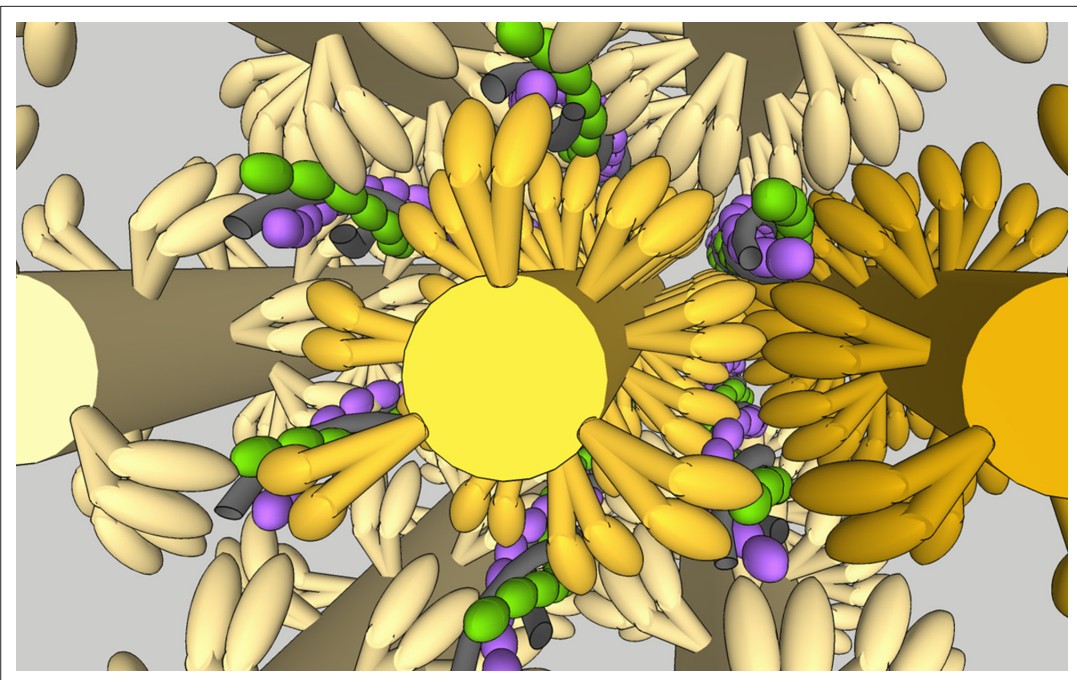

**Figure 6.** Pictorial representation of a sarcomere. The very high density of protein within a sarcomere is clear from this image constructed from recent cryo-electron tomography (cryo-ET) measurements of the positions of the sarcomeric components (*Burbaum et al., 2020*; *Wang et al., 2021*). All components are to scale; thick filaments are shown in yellow, actin filaments in purple/green, and tropomyosin in grey. The reach of the two heads could span between thin filaments. The challenge presented by the close proximity of myosin and actin to relaxation of contraction is immediately apparent. Image is produced in Google Sketchup Make 2017 (data available in *Figure 6—source data 1*).

The online version of this article includes the following figure supplement(s) for figure 6:

**Source data 1.** Google Sketchup file used to create *Figure 6*.

densely packed heads around the thin filament (*Burbaum et al., 2020*; *Wang et al., 2021*; *Figure 6*), it is highly possible for multiple heads to bind in close proximity. Therefore, catastrophic collapse could potentially be mediated over a short distance. This would mean that the tropomyosin's stiffness is likely important for its ability to remove myosins. Consequently, mutations that affect tropomyosin stiffness (e.g., the hypertrophic cardiomyopathy (HCM) mutants E180G and D175N lower tropomyosin stiffness; *Xe et al., 2012*) may mediate their effects through modulating catastrophic collapse. We have previously observed an active region length of ~3 actin pseudo-repeats using laser tweezers (*Kad et al., 2005*), and in this study, we also observed sudden collapses of activation. These may have been indications of the catastrophic collapse we observed here. Since that study used laser tweezers, the thin filament experienced stress on the order of 4 pN. This level of force has been used routinely in the acto-myosin laser tweezers field to remove compliance from the bead-actin-bead assembly (*Finer et al., 1994*; *Dupuis et al., 1997*). Here, there could also be a force present since the tightropes are assembled under flow. Using a similar architecture, but with DNA instead of actin, we have previously measured a tension of 2.2 pN (*Simons et al., 2015*). We expect that the thin filaments would experience a lower tension because of their much greater persistence length (~20 μm at low calcium; *Isambert et al., 1995*), which keeps the thin filaments as rods in solution decreasing the effects of flow extension seen for DNA (*Kad et al., 2010*). It will be interesting to pursue the contribution of thin filament tension to myosin recruitment, since this may also contribute to stretch activation (*Steiger, 1977*). Finally, it is important to consider the cross-talk between adjacent tropomyosins across the actin filament. Recent studies have provided structural evidence (*Yamada et al., 2020*; *Risi et al., 2021*) for earlier reports (*Jewell and Wilkie, 1960*) of cross-talk, meditated through troponin. Our data show clear sparse active regions, consistent with microns of space between clusters. This

suggests that in the conditions used here, it would be unlikely that we are observing binding to opposite sides of the actin filament, indicating collapse is mediated by only a single Tm filament.

## Conclusion

In this study, we have investigated the behaviour of myosin S1 molecules binding metastably to RTFs using a single-molecule imaging approach, offering new insights into how the thin filament relaxes. From our results, it is evident that active regions are turned off in a concerted fashion, a mechanism we term catastrophic collapse mediated by tropomyosin and not requiring the presence of troponin. The role of this 'catastrophic collapse' in the process of muscle contraction remains to be determined, although this nuanced understanding of relaxation may help in our understanding of diseases. HCM is characterised by a deficit in relaxation (*Tardiff, 2005*; *Bai et al., 2011*; *Moore et al., 2012*; *Sewanan et al., 2016*; *Sen-Chowdhry et al., 2016*) that is even evident in the first images of muscle cross-sections where glycogen granules were noted to be absent (*Teare, 1958*). Mutations that reduce the stiffness of tropomyosin (*Xe et al., 2012*) could impair catastrophic collapse, providing a molecular explanation for incomplete relaxation during HCM. In such cases, the thin filament becomes less able to release the myosin heads and completely relax.

## Materials and methods

### Proteins

Myosin and actin were prepared from chicken pectoralis following the protocol of *Pardee and Spudich, 1982*. Both human tropomyosin and human troponin were bacterially expressed in *Escherichia coli*. Tropomyosin with an N-terminal Ala-Ser modification (to mimic acetylation; *Monteiro et al., 1994*) was purified as follows: cell lysate was heated to 80 °C for 10 min before centrifuging at 17,300 ×*g* for 30 min at 4 °C. The supernatant was aspirated and its pH lowered to 4.8, causing Tm to precipitate. After centrifugation at 3000 ×*g* for 10 min at 4 °C, the pellet was resuspended in 5 mM potassium phosphate, pH 7, 100 mM NaCl, and 5 mM $MgCl_2$ before further purification using a HiTrap Q HP column (Cytiva). This procedure was repeated once more and the pure protein was flash frozen in liquid nitrogen and stored in 20 mM Tris, pH 7, 100 mM KCl, and 5 mM $MgCl_2$ at –80 °C.

The troponin complex was purified according to *Janco et al., 2012*. Briefly, cell pellets for each of Troponin I (TnI), Troponin C (TnC), and Troponin T (TnT) were resuspended in 6 M urea, 25 mM Tris, pH 7, 200 mM NaCl, 1 mM ethylenediaminetetraacetic acid (EDTA), 20 % sucrose, and 0.1 % Triton X-100 and sonicated. Following centrifugation at 17,300 ×*g* for 30 min at 4 °C, the supernatants were combined and subsequently dialysed into 2 M urea, 10 mM imidazole, pH 7, 1 M KCl, and 1 mM dithiothreitol (DTT) for 5 hr at 4 °C. To allow the proteins to refold, the urea was removed by a second dialysis into 10 mM imidazole, pH 7, 0.75 M KCl, and 1 mM DTT overnight at 4 °C before a final dialysis into 10 mM imidazole, pH 7, 0.5 M KCl, and 1 mM DTT for 5 hr at 4 °C. The samples were then centrifuged at 17,000 ×*g* for 10 min at 4 °C and the protein in the supernatant precipitated by adding 30 % ammonium sulphate $((NH_4)_2SO_4)$ and gently stirring at 4 °C for 1 hr. After centrifugation at 8000 ×*g* for 30 min at 4 °C, the sample was precipitated on ice for 30 min by bringing the $(NH_4)_2SO_4$ concentration to 50 %. The solution was centrifuged at 17,000 ×*g* for 10 min at 4 °C, and the pellet was resuspended in 10 mM imidazole, pH 7, 200 mM NaCl, 100 µM $CaCl_2$, and 1 mM DTT before an overnight dialysis in the same buffer to remove the $(NH_4)_2SO_4$. After dialysis, the sample was spin concentrated in a 10 kDa molecular weight cutoff (MWCO) column (Amicon Ultra) at 2360 ×*g* for 45 min and further purified using a Sephacryl S-300 column (Cytiva). Finally, the protein sample was stored in the same buffer, with the addition of 3 % sucrose, at –80 °C. Reconstituted RTFs, that is, actin fully decorated with Tm and Tn, were obtained through an overnight incubation at 4 °C in reconstitution buffer as described previously (*Homsher et al., 1996*), with an actin:Tm:Tn ratio of 2:0.5:0.25.

S1-GFP was prepared as described by *Desai et al., 2015*. The S1 region of the myosin heavy chain was digested using 1 mg/ml papain (Sigma-Aldrich) in 5 mM cysteine, pH 6, and 2 mM EDTA for 15 min at room temperature (*Margossian and Lowey, 1982*). The reaction was stopped with 5 mM iodoacetic acid and purified using a DEAE FF column (Cytiva). Regulatory light chain (RLC) C-terminally fused with enhanced GFP and a His-tag (RLC-GFP-6xHis) was recombinantly expressed in *E. coli* and purified using a nickel-nitrilotriacetic acid (Ni-NTA) column. The purified construct was then exchanged for myosin S1 RLC (1:3 S1:RLC-GFP-6xHis ratio) in 50 mM potassium phosphate buffer, pH

7, 600 mM KCl, 10 mM EDTA, 2.5 mM ethylene glycol tetraacetic acid (EGTA), and 2 mM ATP for 1 hr at 30 °C. The reaction was stopped with 15 mM MgCl$_2$ and the construct further purified using a Sephacryl S-200 and a HisTrap HP (Cytiva). Following purification, the ratio of the absorbance at 280 nm and 488 nm was used to ensure that the material was 100 % exchanged. Purified proteins were then stored in 50 % glycerol and 3 % sucrose at –20 °C.

## Creating and imaging thin filament tightropes

RTFs were suspended between 5 µm silica beads (MicroSil microspheres; Bang Laboratories) coated in poly-L-lysine adhered to coverslip of a microfluidic flow cell (*Figure 1a*), as detailed in *Springall et al., 2016*. These RTF tightropes were illuminated using a continuous wave variable-power 20 mW 488 nm diode pumped solid state (DPSS) laser (JDSU), focused off-centre at the back-focal plane of a ×100 objective (1.45 NA) to generate an obliquely angled field (*Kad et al., 2010*; *Tokunaga et al., 2008*; *Konopka and Bednarek, 2008*), custom-built into an Olympus IX50 microscope. Fluorescence was measured using a Hamamatsu OrcaFlash 4.0 camera, observing the interaction of S1-GFP with RTFs in 25 mM imidazole, pH 7.4, 25 mM KCl, 4 mM MgCl$_2$, 1 mM EGTA, and 10 mM DTT. The calcium and S1 concentrations used are given in the figure legends; when multiple concentrations of S1 were used, these data were pooled. We have shown previously that cluster size is affected by [S1]; however, we observed no correlation in collapse probability with [S1]. To reduce background noise from surface-adhered S1-GFPs, the surface of the flow-cell was photobleached for 1 min at 20 mW prior to recording data. Movies were recorded at 3.3 frames/s for 2 min, exciting the GFP with 5 mW laser power, resulting in fluorescent molecules where the S1-GFP co-localises with an RTF (*Figure 1b*). To further improve the signal/noise ratio, the movies were binned 2 × 2, resulting in a pixel size of 126.4 nm.

## Data analysis

### Modelling detachment and attachment probabilities to determine the predicted transition matrix

ATP binding to an attached S1-GFP leads to its detachment. The probability of detachment can therefore be expressed as a function of ATP association:

$$P_d = 1 - e^{-k_T * [ATP] t}$$

where $k_T$ is the second-order ATP-binding rate constant and $t$, the time. Therefore, the probability of $n$ myosins detaching at the same time will be the $n^{th}$ power of $P_d$ multiplied by a scaling factor to account for all the possible combinations of molecules:

$$P_{d(n)} = \frac{P_d^n * i!}{j! n!}$$

where and $j$ represent the starting and final cluster size, respectively.

Likewise, the probability of a myosin joining an already active region can be calculated using the $n^{th}$ power of the attachment probability, $P_a$, leading to:

$$P_{a(n)} = \langle P_a \rangle^n$$

Considering the nature of our assay, $P_a$ cannot be modelled reliably; therefore, we extrapolated it from our data by calculating the mean of the transition rates where only one myosin is seen joining an active region, that is, the values of the first transition to the right of the transition matrix diagonal, resulting in a $\langle P_a \rangle$ of 0.15.

## Reversible Jump Markov Chain Monte Carlo for mixture analysis

As a secondary means to confirm the existence of collapse events, and to ensure that no bias exists in the analysis, we applied a statistical approach to analyse the original rolling-ball-subtracted kymographs. This approach used a Gaussian mixtures model via a Reversible Jump Markov Chain Monte Carlo (RJMCMC) algorithm, based on a custom-written software in R that utilizes the package miscF (*Feng, 2016*) and coda (*Plummer et al., 2010*). The results of the mixture model analysis were used

**Table 1.** The mean, variance, and weight of pixel intensity for a maximum of six or eight myosin binders.

| | Total of six binders (seven components) | | | Total of eight binders (nine components) | | |
|---|---|---|---|---|---|---|
| | Mean = $\mu_6$ | Variance = $\sigma_6$ | Weight = $w_6$ | Mean = $\mu_8$ | Variance = $\sigma_8$ | Weight = $w_8$ |
| Zero binders | 57.17 | 34.73 | 0.12 | 24.52 | 35.58 | 0.10 |
| One binder | 150.75 | 42.07 | 0.21 | 138.57 | 41.54 | 0.15 |
| Two binders | 198.26 | 50.68 | 0.20 | 173.87 | 46.04 | 0.16 |
| Three binders | 254.73 | 60.71 | 0.15 | 210.90 | 51.35 | 0.16 |
| Four binders | 314.23 | 69.58 | 0.14 | 264.22 | 55.51 | 0.14 |
| Five binders | 401.37 | 85.73 | 0.11 | 322.20 | 62.04 | 0.11 |
| Six binders | 551.86 | 111.40 | 0.07 | 382.67 | 80.58 | 0.08 |
| Seven binders | - | - | - | 490.73 | 72.01 | 0.05 |
| Eight binders | - | - | - | 652.38 | 91.13 | 0.04 |

as above to extract the transition table information for the stochastic process of myosin's attachment and detachment.

## Defining transition thresholds from RJMCMC analysis

As a secondary measure to quantify the binding and release of S1-GFP within active regions, we applied a Bayesian statistics approach via RJMCMC. The first step in our analysis was to identify and assign a peak intensity for each point in time for every active region in a kymograph. This was achieved using a custom MATLAB-based fitting routine that uses a Gaussian fit centred on the active region to provide a peak intensity (see 'Materials and methods'). The next step was to convert this intensity into the number of S1-GFP molecules bound. To do this, for one kymograph, we generated four independent Markov chains using the RJMCMC algorithm, and from this, we calculated the average mean pixel intensity values associated with that number of binders. This value depends on the maximum number of fluorescence states (components) used in the RJMCMC algorithm. Since the fluorescence intensity is linear with the number of binders, this was used as a prior constraint for the RJMCMC algorithm.

The average of four simulated RJMCMC chains with two different numbers of components is shown in *Table 1*. For seven components (equivalent to background plus six binders) and nine components (equivalent to background plus eight binders), clear linear increments between binders were determined. However, the nine-component simulation provides a more saturated Gaussian mixtures model. Further confirmation of the choice of component number derives from the weighting or relative abundance of these binders. For nine components, the predominant population is between one and five molecules per bound cluster, but there is still a 17 % chance for six to eight binders (8 % weighting for six binders, 5 % for seven binders, and 4 % for eight binders). Therefore, the values from the nine-component simulation were used to define the fluorescence intensities for the subsequent analysis.

From the values in *Table 1*, we took the simplest approach for choosing the cut-off pixel intensity value of each population of binders by using the mid-point between successive means for each of the nine components. For example, the cut-off value for pixel intensity from baseline to binder 1 (values in *Table 1*: 24.52–138.57) is 81.55 and between binders 1 and 2 (values in *Table 1*: 138.57–173.87) is 156.21; therefore, any peaks in the kymograph within this intensity range would be assigned as a single bound myosin. The mean (*Table 1*) and cut-off values (*Table 1*) were used for analysis of all subsequent kymographs to convert the intensity values of fitted peaks in the data to the number of bound myosins.

## Mixtures analysis

We present a full Bayesian analysis of finite mixtures of univariate normal distributions with an unknown number of clusters. The size of the clusters refers to the number of bound S1-GFP molecules within

that active region. Each observation in the dataset is assumed to have emerged from one of the K clusters. Thus, the purpose of the mixtures analysis is to infer the number (K) of components, the component parameters, and the proportion of each cluster.

RJMCMC methods are adopted to determine the number of clusters in an active region. In this manuscript, the RJMCMC algorithm is employed similarly to *Richardson and Green, 1997*. This method allows the Markov chain to move between the parameter subspaces corresponding to a statistical mixture model with unknown number of components, providing effective model selection and generating a good mixing of the Markov chain. These results were then used to determine the range of values for each component used to transform the raw intensity data into the number of myosins. Using MCMC enables the simultaneous exploration of both the parameters and model space by treating the number of components as a random variable, while being automatically adapted at each step. The RJMCMC analysis regularly proposes a move to a different dimension and rejects this proposal with appropriate probability to ensure the chain crosses the stationary distribution.

The hierarchical model employed in this paper has been proved to be weakly informative by *Richardson and Green, 1997*. This case is more appropriate for our datasets because a prior objective is preferred. The inference should be done mostly based on the data available, as the prior information is not very solid at this stage. Thus, the hierarchical model with fixed α and random β applied for the variance distribution allows a low degree of information to be passed on to the results of the analysis. Following *Richardson and Green, 1997*, we have chosen $\alpha = 2$; g = 0.2; and h = $10/R^2$, where R is the range of the observations.

The results from the RJMCMC analysis qualitatively agree with those reported using the Trackmate method (see *Figure 4—figure supplement 1*), although a higher propensity for collapse is seen. Nonetheless, both methods predict a much-greater-than-expected release probability of multiple myosins.

## Acknowledgements

We thank Rama Desai for collecting preliminary data for this project and for advice. We thank Mike Geeves (University of Kent) for reading the manuscript and also the Kad lab members for useful discussions. This project was funded by the British Heart Foundation (FS/13/69/30504). The authors declare no conflict of interest.

## Additional information

### Funding

| Funder | Grant reference number | Author |
| --- | --- | --- |
| British Heart Foundation | FS/13/69/30504 | Neil M Kad<br>Alessio V Inchingolo |

The funders had no role in study design, data collection and interpretation, or the decision to submit the work for publication.

### Author contributions

Quentin M Smith, Formal analysis, Investigation, Writing – review and editing; Alessio V Inchingolo, Conceptualization, Investigation, Writing – original draft, Writing – review and editing; Madalina-Daniela Mihailescu, Formal analysis, Writing – original draft; Hongsheng Dai, Formal analysis, Methodology, Resources, Supervision, Writing – original draft; Neil M Kad, Conceptualization, Formal analysis, Funding acquisition, Project administration, Software, Supervision, Validation, Writing – original draft, Writing – review and editing

### Author ORCIDs
Quentin M Smith (iD) http://orcid.org/0000-0003-2967-9240
Neil M Kad (iD) http://orcid.org/0000-0002-3491-8595

### Decision letter and Author response
Decision letter https://doi.org/10.7554/eLife.69184.sa1

Author response https://doi.org/10.7554/eLife.69184.sa2

---

## Additional files

### Supplementary files
• Transparent reporting form

### Data availability
All data generated or analysed during this study are included in the manuscript and supporting files.

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
