## [Decision Letter]

**Acceptance summary:**

Your work highlights the complexity of myosin binding and release from thin filaments. The mechanism of concerted release is both important from a molecular point of view, as it illustrates how important it is to consider cooperative effects both in terms of binding and unbinding to actin filaments, and physiologically for all its implications in further understanding muscle contraction. We hope that you will be able to provide in the future a quantitative molecular model accounting for all the subtleties of this mechanism.

**Decision letter after peer review:**

Thank you for submitting your article "Single molecule imaging reveals the concerted release of myosin from regulated thin filaments" for consideration by *eLife*. Your article has been reviewed by 3 peer reviewers, one of whom is a member of our Board of Reviewing Editors, and the evaluation has been overseen by Anna Akhmanova as the Senior Editor. The following individual involved in review of your submission has agreed to reveal their identity: Josh Baker (Reviewer #3).

Essential revisions:

1. The observation that clusters form with tropomyosin but without troponin is curious and questions about the function of troponin and calcium in this system. Does tropomyosin alone inhibit actin-myosin binding? If not, what is the mechanism by which clusters form with only tropomyosin? That is, if tropomyosin alone does not inhibit actin-myosin binding, then how can myosin heads cooperatively activate specific regions?

The authors should provide a kinetic scheme for actin myosin detachment both with and without tropomyosin. This would allow the reader and possibly the authors to better explore what is going on. Specifically, the results seem to imply a detachment rate that depends on the number of bound myosin heads in a cluster. The observation that detachment of the last one or two heads is relatively slow suggests that there is an N-dependent relationship. Is it possible to develop an equation for that N-dependent rate? This would provide significant insights into the mechanism for cooperative detachment. Could such model also explain why the data show only myosin head detachment and not rebinding during the detachment of clusters?

2. The Methods section would benefit from more experimental details. Could the authors specify precisely buffer/protein concentrations in each experiments? For example, it is not clear in Figure 3A whether data at 5 and 10 nM S1-GFP were pulled together or used in different experimental conditions. Is there an influence of myosin-S1 concentration for cluster formation and detachment? What is the rate of photobleaching in these experiments and is it comparable to the rate (not estimated but should be) of detachment?

It is also difficult to assess the precision of the measurements from the few images provided in this work. It would be useful if the authors could detail how the data are processed from typical unmodified images (i.e. detail every step of the data analysis). It would also require examples of how quantitative information is extracted, show individual measurements and not only straight lines as in Figure 2b. For example, the authors should indicate how transient binding of one molecule is distinguished from typical background. The authors should also clarify whether they included every events in their analysis or excluded some weak signals. Such analysis should also include a notion of error, i.e. the degree of confidence when detecting x GFPs in this system, and how such precision impacts the precision of the final transition probabilities.

3. The choice of using GFP-tagged proteins in this system is surprising because these fluorophores are not the brightest for single molecule imaging. Moreover, GFP maturation is slow, which results usually in an important fraction of the proteins being non-fluorescent. Could the authors evaluate by spectrophotometry the fraction of fluorescent vs. non-fluorescent S1-GFP molecules, and tell us how non-fluorescent GFP-S1 contribute to the "real-size" of clusters. How does this change probability values measured in Figures 3 and 4?

4. All reviewers were curious to know whether release of myosin would be altered if the mechanical properties of the thin filaments were modulated. Such experiments are probably too complicated to perform for this specific paper, but could the authors discuss this point and provide estimates of filament tension in these experiments?*Reviewer #1:*

In this paper, the authors use single molecule imaging to reveal how clusters of myosin S1 unbind from reconstituted thin filaments. One of the difficulty of this work was to find experimental conditions where cluster formation is only transient. Data quantification include transition matrices, showing how clusters change in size at these experimental time intervals. These results are compared to theoretical predictions, where the probability of detachment of each molecule is calculated from the parameters that govern attachment time, ADP release and ATP binding. Discrepancies between experimental and theoretical values reveal a mechanism of cooperative disassembly, i.e. where myosin S1 molecules detach by groups and not progressively as expected. However, this theoretical calculation does not explain the mechanism of these "catastrophic" detachments, and only possible explanations are provided in the discussion.

*Reviewer #2:*

Overview: The article authored by Smith et al., describes the importance of understanding how myosin binds and is released from thin filaments with implications in further understanding muscle contraction. To determine the binding interactions of myosin with thin filaments, the authors incorporate single molecule fluorescence imaging in addition to using silica beads to produce attachment sites for the thin filaments. The beads and thin filament complex generates thin filament tightropes which act as key locations for the binding of myosin as they are passed through the flow chamber. From the data, the authors were able to determine the differences between myosin binding and the differing mechanisms of cluster catastrophic collapse and stochastic detachment of myosin. Interestingly, from the imaging data they can show the probabilities for the final cluster sizes dependent on their experimental results and compare to their own predictive model. Overall, this article explores how determining the attachment and detachment of myosin's to thin filaments adds an interesting way to visualize and quantify the active interactions of myosin binding with thin filaments.

The conclusions of this work are mostly supported by the data presented, however, the discussion could be further extended.

*Reviewer #3:*

The goal of this paper is to better understand the molecular mechanisms underlying relaxation of striated muscle. On the surface, this seems like a relatively simple process; however, the results of this study suggest that it is more interesting and significant than this reviewer originally thought. To determine the mechanism of relaxation, the authors use a remarkable technique that allows for the direct observation of myosin binding to actin (the mechanism of contraction) and myosin heads detaching from actin (the mechanism of relaxation). Unlike in most kinetic assay that average binding kinetics both spatially and temporally, this assay reveals binding events at specific locations along an actin filament and detachment events from specific clusters of myosin heads. Thus, the authors were able to observe clusters of myosin heads that bind to actin at specific regions and detachment of these clusters at a rate that exceeds that of a single myosin head, implying cooperative mechanisms. The authors were not able to account for the cooperative detachment mechanism, but for this reviewer that is more intriguing than it is disappointing.

The authors successfully achieved their goal and discussed the significant implications of their results for muscle relaxation in normal and disease states.

---

## [Author Response]

Essential revisions:1. The observation that clusters form with tropomyosin but without troponin is curious and questions about the function of troponin and calcium in this system. Does tropomyosin alone inhibit actin-myosin binding? If not, what is the mechanism by which clusters form with only tropomyosin? That is, if tropomyosin alone does not inhibit actin-myosin binding, then how can myosin heads cooperatively activate specific regions?The authors should provide a kinetic scheme for actin myosin detachment both with and without tropomyosin. This would allow the reader and possibly the authors to better explore what is going on. Specifically, the results seem to imply a detachment rate that depends on the number of bound myosin heads in a cluster. The observation that detachment of the last one or two heads is relatively slow suggests that there is an N-dependent relationship. Is it possible to develop an equation for that N-dependent rate? This would provide significant insights into the mechanism for cooperative detachment. Could such model also explain why the data show only myosin head detachment and not rebinding during the detachment of clusters?

Tropomyosin alone does inhibit myosin’s interaction with actin. This is evidenced from the literature where it has been shown that tropomyosin impedes the motion of actin filaments in the in vitro motility assay (1). In our assay this is also evident from the reduced interactions of myosin with the tightropes in the presence of Tm only, relative to naked actin (2). We have made a statement in the revised version clarifying this point (see end of Results section). Therefore, given that tropomyosin does inhibit interaction with actin then this provides the mechanism for cooperative activation. This has also been shown for the effects of Tm alone on myosin’s ATPase and binding, which we mention both in the Results section and in the discussion.

We thank the reviewers for suggesting we create a model, as a result we have now included a sub-panel in figure 4 containing a cartoon of our interpretation of the results. The principle of catastrophic detachment leads to the release of all heads in a single step. However, without having a clear measure of the detachment rates we feel it is too speculative to suggest an N-head dependent model for detachment. This is a really important measurement that we aim to discover in future studies. Therefore, we have made reference to the importance of discovering if the rate of detachment depends on the number of attached heads, and added a prediction for those values (see discussion).

The last point refers to the attachment and detachment of heads. Our data does reveal both, as seen in the transition tables where association lies to the right of the diagonal. These tables show what happens in a transition, in the case of multiple events we are unable to resolve these due to the time resolution of our measurements, however the probability of these events occurring forms the basis of our interpretation throughout the manuscript.

2. The Methods section would benefit from more experimental details. Could the authors specify precisely buffer/protein concentrations in each experiment? For example, it is not clear in Figure 3A whether data at 5 and 10 nM S1-GFP were pulled together or used in different experimental conditions. Is there an influence of myosin-S1 concentration for cluster formation and detachment? What is the rate of photobleaching in these experiments and is it comparable to the rate (not estimated but should be) of detachment?It is also difficult to assess the precision of the measurements from the few images provided in this work. It would be useful if the authors could detail how the data are processed from typical unmodified images (i.e. detail every step of the data analysis). It would also require examples of how quantitative information is extracted, show individual measurements and not only straight lines as in Figure 2b. For example, the authors should indicate how transient binding of one molecule is distinguished from typical background. The authors should also clarify whether they included every events in their analysis or excluded some weak signals. Such analysis should also include a notion of error, i.e. the degree of confidence when detecting x GFPs in this system, and how such precision impacts the precision of the final transition probabilities.

We apologize for not organizing the details of the method better. The data in Figure 3 was obtained at different S1-GFP concentrations as mentioned in the methods and figure legend, however we were not clear that these data were pooled. This was necessary because the concentration of S1-GFP was tuned for every experiment until we saw metastability. We have now added “*and pooled for analysis”* into the legend to make our approach clearer. No behavioral correlation for the collapses was found with [S1-GFP], as expected for a process that is not concentration dependent. However, cluster size is affected by the concentration of myosin, leading to larger clusters with higher [S1-GFP]. We published a comprehensive analysis of this behavior previously (2), and we have amended the Methods section to make this clear.

With regards photobleaching, we did not carry out a measurement of photobleaching because this is an unreliable measurement to make on tightropes, since their height above the surface is not controlled. We could calculate the photobleaching rate on the surface but the laser power density is stronger. Therefore, we altered the laser power and found S1-GFP lifetimes were unchanged, and we also characterized the second order ATP binding rate constant as previously (Desai et al., 2015) and found it agrees with solution studies. Moreover, the effects of photobleaching cannot explain the collapse that we see here, because photobleaching is also stochastic. We have mentioned these aspects of photobleaching in the manuscript (lines 135-137 and 270).

In response to the request for more analytical details we have created a new figure detailing the steps in the analysis using a second dataset. We highlight the inclusion criteria and indicate estimates on the errors in determining the number of bound myosins in the figure legend. The impact of our confidence in assigning the number of myosins is also discussed (see Results section starting at line 134-159). Details of this analysis have been moved out of the methods and into the results to fit the paper narrative more clearly. Most of this analysis, including the assignments on numbers of myosins has been previously published (2), and the programs used to analyze the data have all been made available on GitHub, linked in the manuscript.

3. The choice of using GFP-tagged proteins in this system is surprising because these fluorophores are not the brightest for single molecule imaging. Moreover, GFP maturation is slow, which results usually in an important fraction of the proteins being non-fluorescent. Could the authors evaluate by spectrophotometry the fraction of fluorescent vs. non-fluorescent S1-GFP molecules, and tell us how non-fluorescent GFP-S1 contribute to the "real-size" of clusters. How does this change probability values measured in Figures 3 and 4?

We agree with the reviewer that GFP is not the brightest fluorophore and we have begun incorporating a different fluorescent protein onto the RLC, however GFP is very good at ensuring enough expressed protein remains in solution. RLC itself forms inclusion bodies, therefore our expression approach biases the RLC-GFP into solution, as much as possible. The process of expressing with another fluorescent protein involves re-optimizing the best procedure for protein production.

With regards the maturation time, the RLC-GFP is purified from bacteria before exchange with the endogenous RLC from myosin S1. Going from expression to use in the assay minimally takes days to complete, and therefore the GFP has ample time to mature. The published maturation τ_90_ times for the GFP we use here are ~90min (3) and maturation yields have been assessed to approach 100% (4). We routinely measure the OD280/OD488 ratio for our proteins following exchange, which also involves two purification steps to separate from unlabeled protein. Our data show 100% fluorophore labelling. Based on this the cluster sizes we determine here are accurate measures. We thank the reviewers for raising this point and have now included a statement in the Methods explaining that the labelling is 100%.

4. All reviewers were curious to know whether release of myosin would be altered if the mechanical properties of the thin filaments were modulated. Such experiments are probably too complicated to perform for this specific paper, but could the authors discuss this point and provide estimates of filament tension in these experiments?

This is a wonderful experiment, and one that we wish to pursue in the near future. It is, of course, potentially very relevant to stretch-activated muscle systems, including cardiac. Could tension lead to changes in the thin filament and recruit myosins? Since tropomyosin is a single molecule spanning the whole of the actin filament, it is reasonable to guess that pulling on the ends would change the energy of the whole protein and could possibly change the occupancy of the thin filament. The tension of the thin filament in these experiments is very difficult to determine, we previously determined the tension on a single DNA molecule tightrope as 2.2 pN. This relatively low force is unlikely to affect acto-myosin interactions, and has been used for decades for single molecule myosin measurements (5). That being said, the tension on thin filaments is likely smaller because of the much greater persistence length of actin, which would reduce the end-to-end tension experienced by the molecule. Arguments based on this have been added to the Discussion section.

References

1. VanBuren P, Palmiter KA, and Warshaw DM (1999) Tropomyosin directly modulates actomyosin mechanical performance at the level of a single actin filament. *Proc Natl Acad Sci U S A* 96(22):12488-12493.

2. Desai R, Geeves MA, and Kad NM (2015) Using fluorescent myosin to directly visualize cooperative activation of thin filaments. *J Biol Chem* 290(4):1915-1925.

3. Balleza E, Kim JM, and Cluzel P (2018) Systematic characterization of maturation time of fluorescent proteins in living cells. *Nat Methods* 15(1):47-51.

4. Macdonald PJ, Chen Y, and Mueller JD (2012) Chromophore maturation and fluorescence fluctuation spectroscopy of fluorescent proteins in a cell-free expression system. *Anal Biochem* 421(1):291-298.

5. Dupuis DE, Guilford WH, Wu J, and Warshaw DM (1997) Actin filament mechanics in the laser trap. *J.Muscle Res.Cell Motil.* 18(1):17-30.